# Rethinking the Operation Pattern for Anomaly Detection in Industrial Cyber–Physical Systems

**Zishuai Cheng \***, **Baojiang Cui and Junsong Fu**

School of Cyberspace Security, Beijing University of Posts and Telecommunications, Beijing 100876, China
* Correspondence: chengzishuai@bupt.edu.cn

**Abstract:** Anomaly detection has been proven to be an efficient way to detect malicious behaviour and cyberattacks in industrial cyber–physical systems (ICPSs). However, most detection models are not entirely adapted to the real world as they require intensive computational resources and labelled data and lack interpretability. This study investigated the traffic behaviour of a real coal mine system and proposed improved features to describe its operation pattern. Based on these features, this work combined the basic deterministic finite automaton (DFA) and normal distribution (ND) models to build an unsupervised anomaly detection model, which uses a hierarchical structure to pursue interpretability. To demonstrate its capability, this model was evaluated on real traffic and seven simulated attack types and further compared with nine state-of-the-art works. The evaluation and comparison results show that the proposed method achieved a 99% F1-score and is efficient in detecting sophisticated attacks. Furthermore, it achieved an average 17% increase in precision and a 12% increase in F1-Score compared to previous works. These results confirm the advantages of the proposed method. The work further suggests that future works should investigate operation pattern features rather than pursuing complex algorithms.

**Keywords:** anomaly detection; industrial cyber–physical systems; operation pattern





## 1. Introduction

Industrial cyber–physical systems (ICPSs) enable smart manufacturing by leveraging emerging techniques such as the Internet of Things (IoT) and the Internet of Services (IoS) to bridge the gap between the physical factory floor and the cyber computational space. This has been the primary enabling technology for Industry 4.0 [1]. ICPSs are widely used in many industries that are closely related to the national economy and people's livelihood, such as electricity generation and distribution networks, oil and gas pipelines, and water distribution systems [2,3]. Their critical role makes them attractive targets for attacks such as criminal control, espionage and cyberwar [4].

Industrial control systems (ICSs) are often deployed in isolated locations and rely on this isolation to defend against cyberattacks [5,6]. However, the integration between ICSs and the enterprise Internet breaks down this isolation and exposes ICSs to cyber threats. For instance, malicious control commands can be injected into systems by data interception and tampering, or an adversary could monitor production processes by eavesdropping on the data transmitted through the Internet [7,8].

In recent years, notorious ICPS attacks have benn frequently reported. The incident that occurred in 2000 at Maroochy Water Services in Queensland, Australia, is the first widely recognised ICPS cyberattack. It caused a spill of 264 K gallons of raw sewage to local parks and rivers. The Stuxnet worm attacked Iranian nuclear facilities from 2005 to 2010, resulting in the destruction of nearly 1000 centrifuges. In 2015, the Ukrainian power section experienced serious cyberattacks from BlackEnergy3, resulting in unscheduled power outages of seven 110 KV and twenty-three 35 KV substations for up to three hours. The ransomware NotPetya-ransomware hit many Ukrainian national facilities (e.g., the metro,

banks, newspapers, Chernobyl nuclear power plant) in 2017. These ICPS attacks caused severe damage to national security and people's lives and drew the attention of researchers.

However, defending against ICPS attacks is a challenging task for the following reasons [5,9]: (1) ICPS often include legacy devices due to the lack of replacement options or costly upgrades [10,11]. These legacy devices are usually equipped with limited computational resources. Therefore, updating programs to support additional security features is challenging. (2) Immediate software patching is always unrealistic, as most devices are critical in the sense that the enterprise cannot afford maintenance downtime [12]. Moreover, some programs are burned into read-only memories (ROMs). (3) Some de-facto protocols were originally designed for serial communication and lacked security mechanisms [13]. For instance, the Modbus TCP protocol does not provide integrity and confidentiality protection. (4) Attacks can damage ICPSs by disrupting control processes without requiring malicious commands. For example, the Stuxnet worm used man-in-the-middle (MITM) technology to intercept and manipulate communications between master and slave devices [14]. In this respect, intrusion detection systems (IDSs) failed to detect these attacks because they usually use legitimate function codes.

To address the limitations of IDSs, researchers have proposed using anomaly-based detection methods to identify malicious operations and cyberattacks, because ICPSs' behaviours usually follow consistent patterns [4]. Once an ICPS is built, its topology and communication protocols are generally fixed (at least for a fairly long time), and frequent software patching and updating are unrealistic [8,15]. Master devices repeatedly poll PLCs at a fixed frequency based on automatic polling programmes and issue repeating sequences of commands [13]. Machine learning (ML) algorithms, especially deep learning (DL) algorithms, provide the ability to learn regular patterns from raw data [16] and are widely used to build anomaly detection models. Mozaffari et al. [17] and Pallavi et al. [18] evaluated several ML and DL algorithms and demonstrated their ability to detect ICPS attacks.

However, as concluded in previous works [5,19], ML-based and DL-based detection models are not entirely adapted to real ICPS systems due to ICPSs' limited computational resources. To address this limitation, Liu et al. [20] proposed using an isolation forest to build anomaly detection models due to its proven effectiveness and low computational complexity. Vita et al. [21] combined echo state networks (ESNs) and auto-encoders for detecting anomalies to pursue a trade-off between memory footprint and inference time. However, most ML and DL algorithms face the "black box" problem, which means that these models can not give human-readable causes for classification results [22].

In addition to the learning algorithms, the description of regular patterns is also vital when building anomaly-based detection models and directly affects detection performance, such as the detection accuracy and the recognition of novel or sophisticated attacks. However, the features, especially the operation pattern features, used in current state-of-the-art works are insufficient. They either use the function code sequence, operation time interval or a combination of the two to describe the operation patterns [13,23,24] or use ML and DL algorithms to automatically learn the representation of operation patterns [25,26]. A vulnerability of this insufficiency is that the models lack the ability to detect sophisticated attacks (e.g., eavesdropping and tampering attacks).

This work investigated real ICPS traffic in a coal mine factory. Then, three features, including operation sequence, operation time interval and execution time, were used to represent the operation patterns. Although operation sequence and time interval were proposed in [13,24], this work expanded on them to improve their descriptive ability. Based on the improved features, this work implemented an unsupervised anomaly detection model by combining basic deterministic finite automaton (DFA), CKmeans and normal distribution (ND) algorithms. This model does not require labelled datasets and has good interpretability, which makes it more adaptable to real ICPS applications. Especially its interpretability could help security experts to understand ongoing attacks and react quickly. The evaluation results regarding real traffic and seven simulated attacks show

that this model achieved a 99% F1-Score with a 1.19% false detection rate. The comparison between the results of the proposed model and nine state-of-art works shows that the proposed model is more efficient in detecting sophisticated attacks and achieves 17% and 12% increments in precision and F1-Score. These evaluation results confirm that well-defined operation pattern features could also improve detection performance. Therefore, this work suggests that future researchers should explore ICPSs' nature and use them to build straightforward, interpretative and unsupervised anomaly detection modes rather than pursuing more complex ML or DL algorithms.

The main contributions of this work are listed as follows:

1.  This work improved the features to describe operation patterns and implemented an unsupervised anomaly-based detection model. The evaluation results for real traffic and seven attack types and a comparison of the results with other works confirm the advantages of the improved features. Additionally, this model does not require labelled datasets and has good interpretability.
2.  This work could inspire further studies of the operation patterns of ICPSs and use them to detect malicious behaviours and cyberattacks.

This paper is organised as follows. Section 2 will discuss the related works. Section 3 provides a brief description of the Modbus TCP protocol. The improved operation pattern features and preprocessing methods are shown in Section 4, followed by the details of the detection model in Section 5. The experiment setup, evaluation and a comparison of the results will be presented in Section 6. A discussion follows in Section 7. Section 8 will conclude the paper.

## 2. Related Work

A series of anomaly detection models have been proposed in recent years, especially after the notorious Stuxnet attack. Good features are important when describing normal behaviours and detecting cyberattacks.

### 2.1. Manual Features

In order to learn the normal pattern, Justin et al. [23] proposed converting industrial traffic into 12 features. The evaluation results of these features, with six supervised machine learning algorithms, show the model achieved a good detection performance for binary classification. Recently, Kalech et al. [24] pointed out that ICPS operations repeated themselves within a well-defined timeframe and proposed detecting attacks using temporal patterns. They proposed five feature extraction methods to represent the temporal patterns and used the hidden Markov model (HMM) and an artificial neural network (ANN) to learn patterns. However, their method only used the operation sequence and time interval features and ignored the execution time taken by slave devices to process the received commands. Therefore, their model has limitations in detecting more elaborate attacks, such as the MITM attacks that increase the execution time.

Li et al. [27] proposed modelling the sequential behaviour using out-of-sequence detection algorithm (OSDA) and detecting abnormal operations that conflict with the normal sequence. However, the complex algorithms make it difficult to interpret the classification results. Goldenberg et al. [13] proposed using the DFA to describe the normal operation sequence for each master–slave channel. Each DFA state is a 33-bit symbol (one bit for Q/R, eight bytes for function code, sixteen bits for reference number and eight bits for the bit/word count). However, this model neglected the inter-dependencies between slave devices. Moreover, this work did not consider the operation time interval and the execution time.

In addition to modelling the sequential operation patterns, some works focus on modelling the amount of industrial traffic because they suppose that periodic operations could trigger traffic periodicity. Hao et al. [28] combined the seasonal auto-regressive integration moving average (SARIMA) and long short-term memory (LSTM) to build a time series pattern. Chen et al. [15] used a burst-deterministic finite automaton (DFA)

model to learn the logic and structure of the burst packets. However, only server attacks cause the traffic variations; thus, they are insensitive when detecting attacks.

Another work that uses normal behaviours to detect attacks can be found in [29]. They proposed that the response results reflect the internal representation of SCADA systems, and variants occur in fixed ranges and can be used to detect abnormal states that could be caused by cyberattacks. This work is significantly different from ours because the proposed work focused on detecting malicious operations that could cause abnormal states. Moreover, it could be too late to detect attacks based on abnormal states because the abnormal states only occur after the attacks have had a significant impact.

### 2.2. Automatic Features

To improve the features' ability, some studies prefer to use machine learning methods to choose or combine the raw features, e.g., Kravchik et al. [9] used principal component analysis (PCA) to process the raw features. Vita et al. [21] further proposed combining echo state networks (ESNs) and auto-encoders (ESN-AE) to extract features and detect anomalies in industrial systems. Zhou et al. [30] proposed using combinations of convolution layers and pooling layers to extract features and identify anomalies using a siamese convolutional neural network (SCNN). Nedeljkovic et al. [31] also proposed using CNN to conduct feature extraction and classification tasks. The automatic features achieve a good performance and could save effort in feature extraction; however, these features make the classification results difficult to explain.

### 3. Modbus Protocol

The Modbus protocol has been a de facto protocol in industrial control systems (ICSs), enabling convenient data exchange between devices. Modbus TCP has been further proposed, which combines Modbus with TCP/IP to facilitate data transmission in Ethernet [32]. Modbus TCP uses the client–server model, where master devices (clients) initialise requests to slave devices (servers). Then, the slave devices process received requests and respond with execution results. For backward compatibility with legacy Modbus devices, gateway or bridge devices are usually deployed between Modbus TCP devices and legacy devices to translate Modbus TCP and Modbus protocols.

As shown in Figure 1, the Modbus TCP consists of a Modbus application (MBAP) header and a protocol data unit (PDU). The PDU is similar to the Modbus remote terminal unit (RTU) message, except that Modbus TCP moves the unit ID into the MBAP header. Modbus TCP further ignores the cyclic redundancy check (CRC) value as the IP and TCP layers have checksum fields, which could provide integrity verifications to the transmitted payload. The fields of Modbus TCP are described as follows.

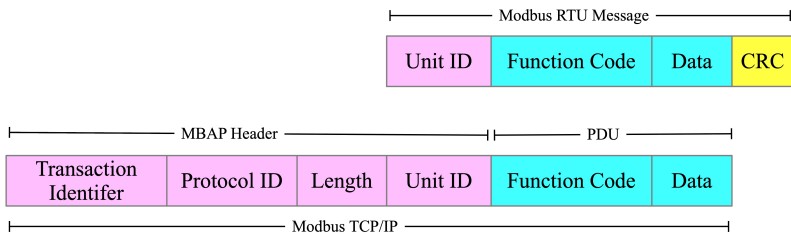

**Figure 1.** The frame of Modbus and Modbus TCP protocols.

- **Transaction Identifier.** This is a two-byte value used to identify Modbus request/response messages. Master devices initialise this value, and slave devices echo it when responding with the execution result.
- **Protocol ID.** This is a two-byte value used to identify the protocol of the following PDU (e.g., 0 for Modbus).
- **Length.** This indicates the byte size of the following data using two bytes.

- **Unit ID (UI).** This is a one-byte value used to help identify slave devices when using gateway/bridge devices. Master devices set UI as the identifier of the requesting slave device. The devices then forward messages to the corresponding slave device according to the UI and set the same UI in their response.
- **Function Code (FC).** FC is a one-byte integer randoming from 1 to 127. The Modbus specification defines the meaning of 19 function codes, such as 2 indicating reading input status and 4 indicating reading registers.
- **Data.** This field usually contains the request parameters or the execution results. Its size is variable, with a maximum limit of 252 bytes. For request commands, this field specifies the start address $Reg_{start}$ and the number $Reg_{cnt}$ of requesting registers. For response messages, it contains the length of the execution results and specific results.

Figure 2 presents an example of the Modbus TCP request and response process, where the master and slave devices are connected by industrial Ethernet. The master device sent a request message with transition ID 20901 to read 13 ($Reg_{cnt}$) discrete inputs (*FC*), starting with 1536 ($Reg_{start}$). The corresponding response message uses the same transition ID, protocol ID and UI as the request. The data in the response message contain the execution result: the content of the 13 discrete inputs is $0 \times 4010$ and the length is 2 bytes.

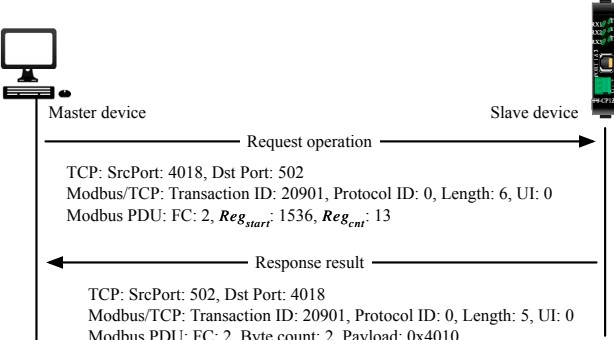

**Figure 2.** An example of Modbus TCP request and response process.

In this work, the MBAP header, function code, request parameters and response results are used to describe operation semantics.

## 4. Features and Preprocessing

In this section, the improved operation pattern features are introduced, followed by the details of data preprocessing.

### 4.1. Operation Pattern Features

To improve the ability to describe ICPSs' normal operation pattern, real industrial traffic collected from a coal mine factory was first analysed. This work then proposed using the operation sequence, time interval and execution time to describe the operation patterns. The first two are similar to the method proposed in [13,24]; however, this work expands on this by improving the description of function types. Some proposals for modelling information systems [33] and performing functional safety assessments [34] can be useful when modelling operation sequences. They proposed using the Markov chain to describe the time-dependent processes; however, this work uses the determined finite state (DFA) because the ICPS operations usually follow a fixed sequence. The execution time is different from the runtime proposed in [35]. The runtime measures the time taken by master devices; however, the execution time focuses on measuring the time taken by slave devices to process the requests.

These improved features are detailed as follows:

- **Operation sequence.** This feature describes the sequence pattern of operation commands, such as the reading register $reg_i$ of device *A*, which is always followed by the

writing register $reg_j$ of device *B*. This feature enables detection models to identify malicious operations that violate the sequence patterns, such as tampering, reconnaissance, and unauthorised operation attacks.

- **Time interval.** This feature focuses on the time intervals between contiguous operations because they often follow fixed distributions [36]. In this way, the detection model is able to detect attacks that occur significantly earlier or later than expected, such as MITM and relay attacks.
- **Execution time.** The execution time describes the time taken by slave devices to process received requests. This feature is reasonable because slave devices are usually real-time systems, and the execution time only depends on the request commands' types and the computational resources [37,38]. This feature enables models to detect anomalies whose responses are earlier or later than expected.

*4.2. Data Preprocessing*

In order to extract features from industrial traffic, this work first reassembles requests and corresponding responses by examining transaction identifiers, followed by extracting features by observing request and response messages. As the ICPS might use gateways when reassembling, this work identifies slave devices using a three-tuple symbol $Addr_{slv}$, which consists of the slaves' IP address $IP_{slv}$, TCP port $Port_{slv}$, and the unit ID *UI*. In the following, the method used to extract features is described.

**Operation sequence-related features.** Previous works used the function code *FC* to describe the meaning of the observed operation [13,24,27]. However, this description is limited because requests that use the same function code to read/write different registers have significantly different meanings in ICPS scenarios. This work addressed this limitation by describing the observed operation type using a three-tuple *FN*, which consists of a function code *FC*, starting address $Reg_{start}$ and the number of accessing registers $Reg_{cnt}$. Therefore, our model learnt not only the function code but also detailed information about the operating registers.

**Time interval-related features.** The method of extracting time interval-related features is similar to the one proposed by Kalech et al. [24]. They proposed two methods to measure the time interval. One measured the elapsed time from the last operation, and the other measured the elapsed time from the last similar operation that has the same *FN*. This work expanded on their extraction method by calculating the time that elapsed when using master–slave**s** channels rather than master–slave channels. This improvement enabled our model to learn the operation sequence information between slave devices. For example, the system operates the gas valve using device *A* only after reading the temperature from the sensor device *B* and takes around 500 ms to calculate the control logic.

**Execution time-related features.** It is difficult to obtain an execution time from response messages because the slave devices do not report this information [17]. This work calculates the execution time by measuring the time that elapsed between the last request packet and the first response packet. This time is measured at the detection system that feeds the mirrored industrial traffic from network switches. We would like to clarify that this is an approximate execution time because it contains transmission latency. In the normal network environment, this latency is unstable and is affected by network quality, e.g., network jitters. However, this calculation is reasonable in ICPS because the transmission latency is very small due to the high real-time requirement of ICPS. For example, according to the electric power systems reference architecture IEC 61850, the maximum transmission latency is 3 ms [11].

In summary, this work presents the following feature improvements. (1) A combination of function code *FC* and the starting and the total number of operation registers $Reg_{start}$ and $Reg_{cnt}$ are used to represent the operation function type, rather than solely using *FC*. (2) The time interval features are learnt using master–slave**s** channels rather using than master–slave channels to learn the inter-dependency between slave devices. (3) The execution time is measured by comparing the request and response packets to enable the

detection model to learn the execution time for different operation types. The proposed features are summarised in Table 1, where the features are structured into four groups. The first is identity-related features, which uniquely identify the master and slave devices. The operation sequence-related feature describes the semantics of each operation, and the time interval-related feature measures the interval between contiguous operations. The final feature is the execution time-related feature, *RT*, which measures the time required by slave devices to process operations.

After feature extraction, the ICPS traffic is represented as a matrix $D = \{op_1, op_2, \cdot, op_n\}$, where $n$ is the number of operations and $op_i$ is the features of the $i$-th operation. We would like to point out that this work could rapidly be extended to other industrial protocols (e.g., Profinet) with minor modifications in protocol decoding.

**Table 1.** Summary of the proposed features. The features are structured into four groups, namely identity, operation sequence, operation time interval and execution time.

| Group | Feature | | Description |
|---|---|---|---|
| Identity-related | | $IP_{mst}$ | The IP address of the master device |
| | $Addr_{slv}$ | $IP_{slv}$ | The IP address of the slave device |
| | | $Port_{slv}$ | The TCP/UDP port |
| | | $UI$ | The unit identity |
| Operation sequence-related | $FN$ | $FC$ | The function code |
| | | $Reg_{start}$ | The start address of the accessing registers |
| | | $Reg_{cnt}$ | The number of accessing registers |
| Operation time interval-related | $ET$ | $ET_{last}$ | The time that elapsed from this operation to the last operation |
| | | $ET_{lastSim}$ | The time that elapsed from this operation to the last similar operation |
| Execution time-related | | $RT$ | The time taken by the slave device to process the received request |

## 5. The Framework of Proposed Unsupervised Anomaly Detection Model

This section first provides the objective of the proposed model, followed by the framework and details. This work focused on building a detection model for systems that only contain one master device and one or more slave devices, particularly the master–slaves channel. If the ICPS has multiple master devices, different detection models are required to describe the operation pattern for each master device because different master devices are usually equipped with different software and have different control functionalities, as well as different operation patterns. The key assumption of this model is that the operations are highly periodic and follow a constant pattern. This assumption is reasonable due to the stability of ICPS [8,15] and automatic control programmes [13].

Most state-of-the-art works prefer to use machine learning (ML) or deep learning (DL) algorithms to build complex models to detect malicious operations and attacks. However, these algorithms usually have a weak interpretability. This makes it challenging for them to output human-readable classification causes. Moreover, in practical scenarios, the raw traffic collected from ICPSs is usually unlabelled, which limits the training of supervised anomaly detection models. Therefore, this work focuses on using unsupervised methods to build an interpretable anomaly detection model.

The framework of the proposed model is shown in Figure 3. This model organises the proposed features into four defence lines. This model has good interpretability because each line uses a specific feature to detect attacks. Specifically, the first defence line is achieved in the feature extraction phase, which focuses on detecting elementary attacks (e.g., the malformed packets and re-transmission) that violate the Modbus TCP standards. The following three defence lines are designed to detect more sophisticated attacks, such as MITM, unauthorised operations and injection attacks, based on the operation sequence, time interval, and execution time features. The details of these defence lines are described as follows.

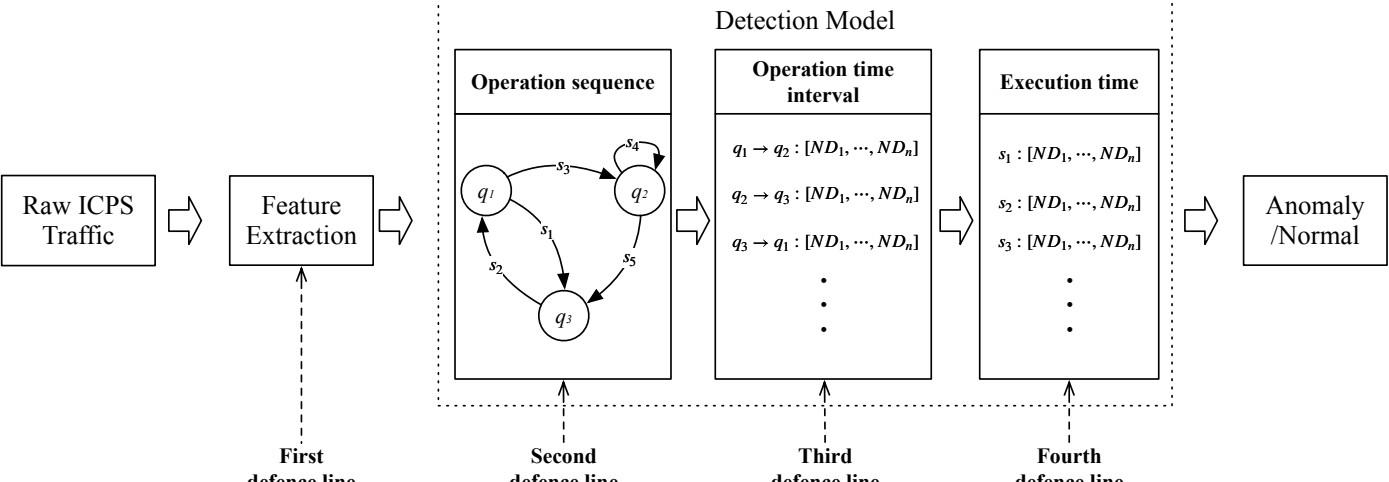

**Figure 3.** The architecture of the proposed unsupervised anomaly detection model. The first line detects elementary attacks at the packet level by examining whether the traffic follows Modbus TCP protocol standards. The last three lines are designed to use the operation sequence, time interval and execution time to detect more advanced attacks. The operation sequence is learnt and described by the deterministic finite automaton (DFA), where nodes $q$ represent states and edges $s$ are the transitions triggered by operations. The operation time interval pattern is learnt on each edge to describe the transition time distribution. The execution time also is learnt on the edges to describe the time taken by slaves to execute this type of operation.

### 5.1. The First Defence Line

This defence line can detect several elementary attacks that violate the Modbus TCP protocol in the data preprocessing phase. These attacks are summarised as follows:

- **Malformed packets.** Packets do not obey the standard protocol implementations, such as the length value not being equal to the actual payload size. These packets are usually observed in fuzzing or manipulating attacks.
- **Missing responses.** Master devices do not receive the expected responses. These anomalies often occur when attackers use selective forwarding to block specific responses.
- **Duplicated responses.** For a request, the master devices receive two or more response messages. These anomalies usually occur when attackers try to disturb the production processes by injecting false responses or invalid control commands [39].
- **Unmatched responses.** The observed responses have no corresponding requests. Many attacks, such as tampering and replay attacks, could trigger this malicious behaviour.

To detect these violations, the model first parses the raw industrial traffic into plain text format according to Modbus TCP protocol. If the parse fails, the model reports these packets as malformed. Then, the model matches the request and response messages according to $IP_{mst}$, $Addr_{slv}$, $FN$ and transaction identifier to examine if the request and response messages are one-to-one mapped. If they are not one-to-one mapped, this defence line reports this abnormality, along with corresponding causes.

Once this defence line identifies packets as malicious, the model directly reports them along with the above-mentioned causes and stops further examinations.

### 5.2. The Second Defence Line

This line aims to detect malicious operations that violate the operation sequence pattern, such as unauthorised operations, unknown master–slave devices and out-of-sequence operations. This section first introduces the method that automatically learns the sequence pattern, followed by the method used to detect anomalies.

**Learn sequence pattern.** This work uses the Deterministic Finite Automaton (DFA) to describe the sequence pattern and examine whether an observed operation is abnormal. This learning approach is similar to the one proposed by Goldenberg et al. [13]. However, this paper extends their method by using the improved operation features and learning operation patterns on the master–slave channels.

The classical DFA is a five-tuple $(Q, \Sigma, \delta, q_0, F)$, where $Q$ is the finite set of states, $\Sigma$ is finite set of input symbols, $\sigma$ is transition function: $Q \times \Sigma \rightarrow Q$, $q_0 \in Q$ is the initial state, and $F \subseteq Q$ are the accept states. Two modifications were made to use the DFA to model operation sequence patterns:

- No initial state $q_0$ and final accept states $F$ are required. ICPS is running in endless repetition, so the end of the operation sequence cannot be observed. As the detection model can be deployed at any time, we cannot expect the first observed state $q_0$ to be fixed. Thus, the state $q_0$ is defined as the first state in the observed operation sequence.
- The input symbol $s \in \Sigma$ is defined as a combination of $Addr_{slv}$ and $FN$. The symbol can be represented as four-tuple $(Addr_{slv}, FC, Reg_{start}, Reg_{cnt})$. For example, a symbol could be *{\*.\*.41.10,520,0,4,0,38}*, where the *\*.\*.41.10* is the slave IP address $IP_{slv}$, *520* is the TCP port $Port_{slv}$, *0* is the unit identity *UI*, *4* is the function code *FC*, *0* is the start address $Reg_{start}$, and *38* is the counter of registers $Reg_{cnt}$.

Therefore, the DFA model used in this work can be represented as three-tuple $(Q, \Sigma, \delta)$.

To model operation patterns, this work first splits the dataset $D$ into master–slave channels according to the masters' addresses. Each channel consists of a unique master device and all connected slave devices. We would like to highlight that a slave device can be included in different channels depending on the ICPS's topology. Therefore, $D$ is further represented as several subsets $\{d_1, \ldots, d_k\}$, as shown in line 3 of Algorithm 1, where $k$ is the number of master devices in the ICPS.

This work then extracts the symbol $s$ of each operation $op$ for each subset $d$ by combining the slave device identifier $Addr_{slv}$ and operation type $FN$. After symbol extraction, each subset is represented as a symbol string, and we then trained the DFA model on each string. Finally, several DFA models $\{DFA_1, \ldots, DFA_k\}$ were learnt to describe the operation sequence patterns, where $DFA_i$ represents the pattern for the $i$-th master–slave channel. The detailed procedure of learning the operation sequence model is presented as the *LearnModel* function in Algorithm 1.

Figure 4 shows an operation sequence example learnt from the coal mine factory, where the node indicates the internal state, and the edge represents the transition from one state to another. In the detection phase, the DFA model determines the transition edge according to the current state $q_{curr}$ and the observed symbol $s$. For example, if the current state is $q_2$ and the observed symbol is $s_3$, the state that is reached should be $q_4$.

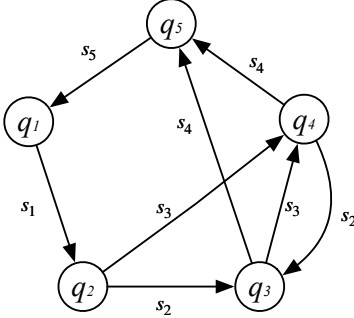

**Figure 4.** An example of an operation sequence pattern represented by the DFA model. The transition edge is determined based on the current state $q_{curr}$ and the observed symbol $s$. The ground-truth operation cycle of this pattern is shown in Table 2.

---

**Algorithm 1:** Operation sequence model

---

**Input:** dataset of operations $D$

**Output:** operation sequence model $M$

1 **Function** LearnModel($L$):

2      $M = \{\}$

3      split $D$ into $\{d_1, \cdots, d_k\}$ according to master address $IP_{mst}$, where $k$ is the number of master devices

4      **for** *each d in* $\{d_1, \cdots, d_k\}$ **do**

5          string $str = \varnothing$

6          **for** *each operation op in d* **do**

7              $IP_{slv}, Port_{slv}, UI, FC, Reg_{start}, Reg_{cnt} \leftarrow op$ ; // extract features

8              construct symbol $s$ using the combination $\{IP_{slv}, Port_{slv}, UI, FC, Reg_{start}, Reg_{cnt}\}$

9              append $s$ into $str$

10          **end**

11          $(Q, \Sigma, \delta) \leftarrow$ DFA($str$)

12          append $(Q, \Sigma, \delta)$ into $M$ with the index $IP_{mst}$

13      **end**

14      **return** $M$

15 **end**



**Input:** observed operation $op$, operation sequence model $M$

**Output:** classification result

16 **Function** DetectionModel($M$, $op$):

17      $IP_{mst} \leftarrow op, s \leftarrow op$ ; // extract features

18      **if** $IP_{mst}$ *in* $M$ **then**

19          $(Q, \Sigma, \delta) \leftarrow M$ ; // retrieve DFA according to $IP_{mst}$

20          $q_{curr}$ ; // current state

21          **if** $s$ *in* $\Sigma$ **then**

22              **if** *exist transition for s according to $\delta$ and $q_{curr}$* **then**

23                  $q_{curr} \leftarrow \delta(q_i, s)$ ; // update state

24                  **return** *normal*

25              **else**

26                  **return** *abnormal* ; // deviation transition

27              **end**

28          **else**

29              **return** *abnormal* ; // unknown symbol

30          **end**

31      **end**

32      **return** *abnormal* ; // no matching master

33 **end**

---

**Detect anomalies.** This defence line identifies malicious operations by examining if they obey the normal operation sequence pattern. The examination is performed according to the following steps. The observed symbol $s$ is first constructed by extracting and combining the slave device identifier $Addr_{slv}$ and function type $FN$. The associated DFA model $DFA_i$ is then retrieved based on the master device's address, and this model is then used to determine if the observed operation is out-of-sequence based on current state $q_{curr}$ and observed symbol $s$. The detailed procedure of examining the observed operations is presented as the *DetectionModel* function in Algorithm 1. The transition results typically fall into one of the following four categories:

- **No matching master.** The master device $IP_{mst}$ has not been observed in the training dataset. This often means attackers are simulating master devices to operate the slave devices to inject malicious commands.
- **Unknown symbol.** The symbol has never been observed in the training dataset. Several attacks, such as reconnaissance and scanning attacks, could generate unknown symbols.
- **Deviation transition.** There is no transition for the observed symbol, or the transmitted state is different from expected. This transition often means the observed operation violates the normal operation sequence though its function type $FN$ is legitimate. Several attacks, such as replay, impersonation, or selective forwarding attacks, could lead to this violation.
- **Normal.** The state transits as expected.

Suppose this defence line determines the observed operation is normal. In this case, it updates its state to the transmitted state and waits to examine the next operation. Otherwise, this defence line directly reports the observed operation with detailed causes and stops further examinations of this operation in the third and fourth defence lines.

*5.3. The Third Defence Line*

The first and second defence lines detect attacks at the packet and operation sequence levels. However, some sophisticated attacks are able to bypass these lines because they will not trigger malformed packets or operation pattern disturbances. The goals and behaviours of these attacks are summarised below:

- **MITM attack.** The attackers' goals are tampering or eavesdropping on the request or response messages without affecting their sequence and function type. Stuxnet is a typical instance of disrupting production processing by modifying the request commands and response results [14]. These attacks usually involve interception and forwarding processes. Therefore, the transmission time significantly increases [40].
- **Resource exhaustion attacks.** These attacks try to slow the processing of master or slave devices by exhausting computational resources (e.g., CPU, memory). Given their goal, the operation time interval or execution time increase.

This work proposed using the operation time interval and execution time to detect these sophisticated attacks. The operation time interval is described in this section, and the execution time is discussed in the next section.

**Learn time interval pattern.** This work uses the normal distribution (ND) model to describe the operation time interval pattern because master devices usually rely on automatic polling procedures to initiate requests. For instance, the system checks the temperature every 5 s [36]. Specifically, ND models are learnt on each transition edge (e.g., the edge from state $q_1$ to $q_2$ in Figure 4) of each DFA model.

The normal distribution is a common rule used to find outliers. This depends on the assumption that the data are normally distributed, which is often not the case. In ICPS, the time intervals on a transition edge could be mixed because the transition could repeat several times, with different intervals, in one operation cycle. For example, the transition from $q_1$ to $q_2$ triggered by the symbol $s_1$ repeats three times at *i + 1,i + 6* and *i + 11*, as shown in Table 2.

To address this challenge, this work proposes learning the operation time interval pattern as follows. (1) CKMeans [41] clustering algorithm is first used to cluster the time interval values into sub-clusters for each transaction. CKMeans is a dynamic programming algorithm for optimal 1D clustering. Similar to the widely used *k*-means algorithm, they partition complex datasets into *k* groups, such that the sum of squared Euclidean distances to each group mean is minimised, but the CKMeans guarantees optimality in the 1D clustering problem. (2) The model then learns the operation time interval pattern on each sub-cluster and describes its pattern using normal distribution $\mathcal{N}(\mu, \sigma^2)$, where the mean $\mu$ is the execution time and deviation $\sigma$ is the noise transmission latency. (3) Finally, the operation time interval pattern for each transition edge is represented as $\{\mathcal{N}_1, \mathcal{N}_2, \ldots, \mathcal{N}_n\}$, where *n* is the number of the sub-clusters for this transition edge. The detailed procedure of learning the time interval model is represented in Algorithm 2.

**Detect anomalies.** This defence line detects malicious operations by examining whether the observed time that elapses satisfies the operation time interval pattern. After the inspection of the second defence line, this line can be made aware of the corresponding transaction edge based on the DFA model. Then, it retrieves the corresponding normal distribution modes based on the transaction edge. Finally, as shown in *DetectionModel* function of Algorithm 2, this line uses the three-sigma rule to determine if the elapsed time satisfies each of these models. The operation is classified as normal if the elapsed time satisfies at least one model. Otherwise, this line reports the observed operation as anomalous with detailed causes and stops further inspection.

---

**Algorithm 2:** Time interval model

---

**Input:** time interval values: $V=\{ET_1, ET_2, \cdots, ET_i\}$
**Output:** time interval model $M$

1 **Function** LearnModel($V$):
2      $M = \{\}$
3      cluster result $C \leftarrow$ CKMeans($V$)
4      **for** *each cluster i in C* **do**
5          $d \leftarrow$ time interval values contained in cluster $i$
6          means $\mu = \frac{\sum_{j=1}^{|d|} d_j}{|d|}$, variance $\delta^2 = \frac{\sum_{j=1}^{|d|}(d_j - \mu)^2}{|d|}$
7          $\mathcal{N}_i \leftarrow$ ND($\mu, \delta^2$)
8          append $\mathcal{N}_i$ into $M$
9      **end**
10      **return** *time interval model M*
11 **end**

**Input:** observed time interval $v$, time interval model $M$
**Output:** classification result

12 **Function** DetectionModel($M, v$):
13      **for** *each model $\mathcal{N}_i$ in M* **do**
14          **if** *v in* $[\mu_i - 3*\delta, \mu_i + 3*\delta]$ **then**
15             **return** *normal*
16          **end**
17      **end**
18      **return** *abnormal*
19 **end**

---

**Table 2.** Operation pattern of the coal mine system based on 170 h' observation.

| No. | $IP_{mst}$ | $Addr_{slv}$ | | | FC | $Reg_{start}$ | $Reg_{cnt}$ | Dist. of $ET_{lst}$(ms) | Dist. of $ET_{lstSim}$(ms) | Dist. of $RT$(ms) |
|---|---|---|---|---|---|---|---|---|---|---|
| | | $IP_{slv}$ | $Port_{slv}$ | UI | | | | | | |
| i + 1 | *.*.204.73 | $(*.*.41.10,$ | $520,$ | $0)$ | 4 | 0 | 38 | $N(687.8, 1.9^2)$ | $N(1073.5, 3.5^2)$ | $N(51.2, 1.9^2)$ |
| i + 2 | *.*.204.73 | $(*.*.41.10,$ | $520,$ | $0)$ | 4 | 1536 | 4 | $N(187.8, 1.7^2)$ | $N(1021.1, 3.2^2)$ | $N(60.6, 1.8^2)$ |
| i + 3 | *.*.204.73 | $(*.*.12.85,$ | $520,$ | $0)$ | 2 | 1536 | 11 | $N(50.1, 1.5^2)$ | $N(1124.0, 3.8^2)$ | $N(310.3, 1.9^2)$ |
| i + 4 | *.*.204.73 | $(*.*.12.85,$ | $520,$ | $0)$ | 4 | 1536 | 2 | $N(124.8, 1.7^2)$ | $N(1133.5, 3.8^2)$ | $N(57.6, 1.9^2)$ |
| i + 5 | *.*.204.73 | $(*.*.41.10,$ | $520,$ | $0)$ | 2 | 1536 | 13 | $N(62.4, 1.5^2)$ | $N(1112.9, 3.8^2)$ | $N(272.3, 1.9^2)$ |
| i + 6 | *.*.204.73 | $(*.*.41.10,$ | $520,$ | $0)$ | 4 | 0 | 38 | $N(812.7, 1.8^2)$ | $N(1237.8, 3.7^2)$ | $N(51.3, 2.0^2)$ |
| i + 7 | *.*.204.73 | $(*.*.41.10,$ | $520,$ | $0)$ | 4 | 1536 | 4 | $N(62.6, 1.5^2)$ | $N(1112.7, 3.6^2)$ | $N(60.5, 1.9^2)$ |
| i + 8 | *.*.204.73 | $(*.*.41.10,$ | $520,$ | $0)$ | 2 | 1536 | 11 | $N(62.9, 1.6^2)$ | $N(1125.4, 3.5^2)$ | $N(310.2, 1.9^2)$ |
| i + 9 | *.*.204.73 | $(*.*.12.85,$ | $520,$ | $0)$ | 4 | 1536 | 2 | $N(62.1, 1.6^2)$ | $N(1062.8, 3.5^2)$ | $N(57.6, 2.0^2)$ |
| i + 10 | *.*.204.73 | $(*.*.12.85,$ | $520,$ | $0)$ | 2 | 1536 | 13 | $N(46.6, 1.6^2)$ | $N(1047.0, 3.7^2)$ | $N(272.5, 1.9^2)$ |
| i + 11 | *.*.204.73 | $(*.*.41.10,$ | $520,$ | $0)$ | 4 | 0 | 38 | $N(750.3, 1.7^2)$ | $N(984.6, 3.7^2)$ | $N(56.2, 2.1^2)$ |
| i + 12 | *.*.204.73 | $(*.*.41.10,$ | $520,$ | $0)$ | 2 | 1536 | 11 | $N(187.4, 1.6^2)$ | $N(1046.5, 3.3^2)$ | $N(310.3, 1.9^2)$ |
| i + 13 | *.*.204.73 | $(*.*.41.10,$ | $520,$ | $0)$ | 4 | 1536 | 4 | $N(52.8, 1.5^2)$ | $N(1162.1, 4.0^2)$ | $N(60.7, 1.9^2)$ |
| i + 14 | *.*.204.73 | $(*.*.12.85,$ | $520,$ | $0)$ | 4 | 1536 | 2 | $N(62.5, 1.1^2)$ | $N(1099.6, 3.4^2)$ | $N(57.8, 1.9^2)$ |
| i + 15 | *.*.204.73 | $(*.*.12.85,$ | $520,$ | $0)$ | 2 | 1536 | 13 | $N(83.1, 1.7^2)$ | $N(1136.0, 3.4^2)$ | $N(272.4, 1.9^2)$ |

*5.4. The Fourth Defence Line*

This defence line is designed to learn the execution time pattern for each operation function type *FN* and use this pattern to detect malicious operations that deviate from the normal range.

**Learn execution time pattern.** The learning approach for this defence line is similar to that of the third line. The difference is that the dataset used in this line shows the execution time of each operation type. This line also uses the normal distribution model to describe execution time patterns and describes the pattern for each operation type as a set of models.

**Detect anomalies.** The detection method is also similar to the approach used in the third defence line. This line relies on the operation type *FN* rather than the transaction edge to retrieve the execution time modes. This means that the operation time interval defence line can only be placed after the operation sequence line; however, this defence line can be placed anywhere after the first line. If the execution time satisfies at least one model, this line classifies it as a normal operation. Otherwise, this defence line reports it as malicious, with detailed causes.

This unsupervised anomaly detection model uses four defence lines to identify abnormal operations and attacks. This model classifies an operation as normal only if it satisfies all defence lines; otherwise, this model reports it as abnormal, with detailed causes.

## 6. Experiments and Results

The Secure Water Treatment (SWAT) dataset [42,43] is the most popular ICPS dataset, and has been widely used to evaluate the performance of anomaly-based detection works. However, this dataset lacks precise timestamps to calculate the time interval and execution time. Power System datasets [23,44], the Electra dataset [45] and the Water Storage Tank dataset [44,46] also lack detailed information to train the proposed model.

In this work, the proposed model was evaluated on a testbed, as shown in Figure 5. This testbed emulated an ICPS subsystem of a coal mining factory using Open-PLC project [47,48]. It consisted of one master device with IP *{\*.\*.204.73}* and two slave devices *{\*.\*.41.10,520,0}* and *{\*.\*.12.85,520,0}*. A laptop installed with Kali was used to launch cyberattacks. Our model ran on another computer and was fed the mirrored traffic. The operation pattern that was run in this testbed is shown in Table 2 and is the same as the ICPS subsystem of the coal mine factory. Each operation cycle consisted of fifteen operations that operated two slave devices with five unique operation types. This ICPS subsystem comprised one master device, two slave devices, and several application servers (e.g., FTP and HTTP servers). A total of 170 h of industrial traffic were collected at the network switch from the factory to analyse the operation pattern, and we only observed the master device operating the slave devices in this collected traffic.

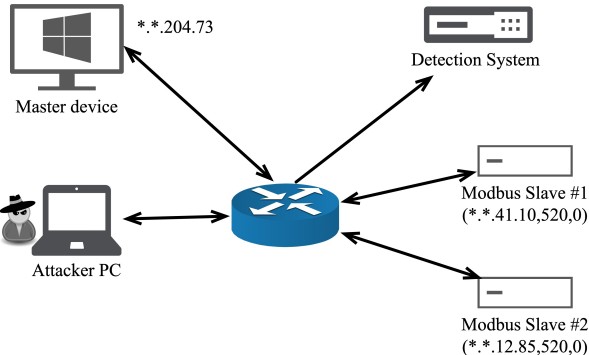

**Figure 5.** The framework of the testbed, which contains one master device, two Modbus slave devices, an attacker PC and the proposed detection model. The network switch connects all components. All packets that go through the switch are mirrored to the anomaly detection system for anomaly detection.

### 6.1. Attack Scenarios

In order to evaluate the proposed model, seven attack scenarios were simulated in the testbed. These were common network-based attacks, targeting the networking part of the ICPSs such as packets, protocols or production processes [49]. This section details these attacks as follows:

- **Eavesdropping attack.** This aims to eavesdrop on production processes using MITM technology. It works as a relay, which forwards packets between master and slave devices and keeps a copy of the traffic to infer production processes.
- **Tampering attack.** This attack is similar to the previous one. The main difference is that attackers modify the operation values. They aim to disturb the production process.
- **Selective forwarding attack.** This attack uses MITM technology to forward certain operation types and drop other types. Its goal is to disturb the production process, as in the tampering attack.
- **Denial-of-Service (DoS) attack.** This tries to slow down or interrupt systems by flooding packets or consuming massive computational resources.

- **Replay attacks.** This attack replays a certain copy of the operations to mislead the control program and disturb the production process.
- **Unauthorised operation.** The operation whose function type *FN* has not previously been observed. This could be caused by the attackers using MITM or malware programs to manipulate slave devices.
- **Unauthorised device.** The master or slave device has never been observed in the training dataset. This anomaly could be caused by an attacker connecting his/her tools to the network.

For eavesdropping, tampering and selective forwarding attack scenarios that rely on the MITM technology to intercept and forward packets, the operation time interval or the response time increases. In this work, this increment was measured as $T_{diff}$. As the last two defence lines rely on this time increment to detect malicious operations, this work controlled the range of $T_{diff}$ to evaluate the model's sensitivity. Specifically, this work used three ranges: $[10, 20)$, $[20, 50)$ and $[50, 100]$. In this work, the man-in-the-middle was used to control the operation time interval and the response time variants, where we controlled the forwarding (processing) latency using the *sleep* command.

This work collected 15,000 benign operations to train the model. A total of 137,000 benign and 15,000 malicious operations were collected for each attack type to evaluate the detection performance.

### 6.2. Performance Metrics

The aim of this paper is to detect ICPS anomalies and attacks based on the operation pattern. To measure the performance of the proposed model, we focused on the metrics of precision (Equation (1)), recall (Equation (2)), F1-score (Equation (3)), and false-positive rate (FPR) (Equation (4)).

$$Precision = \frac{TP}{TP + FP} \tag{1}$$

$$Recall = TPR = \frac{TP}{TP + FN} \tag{2}$$

$$F1\text{-}Score = \frac{2 * Precision * Recall}{Precision + Recall} \tag{3}$$

$$FPR = \frac{FP}{FP + TN} \tag{4}$$

where $TP$ is the number of abnormalities detected as abnormal, $FN$ is the number of abnormalities detected as normal, $TN$ is the amount of normal data detected as normal, and $FP$ is the amount of normal data detected as abnormal. F1-score [50] is defined as the harmonic mean of the precision and recall, which is a widely used metric to evaluate the classification performance in imbalanced data.

### 6.3. Evaluation Results

This section describes the overall detection performance for the above-described seven attack scenarios, followed by a performance analysis of the last two defence lines and a demonstration of interpretability.

6.3.1. Overall Performance.

As shown in Table 3, the proposed model is efficient in detecting selected forwarding, replay, and unauthorised operation/device attacks because these attacks often break the operation sequence pattern. Specifically, the forwarding and replay attacks drop or add request or response messages, and thus break the sequence pattern. Unauthorised operations and unauthorised slave device attacks will trigger the unknown symbol abnormality at the second defence line. This is because the symbol $s$ is a combination of the slave device address and operation type ($Addr_{slv}, FC, Reg_{start}, Reg_{cnt}$); the unauthorised slave devices

and unauthorised operations attacks will affect $Addr_{slv}$, $FC$, $Reg_{start}$ or $Reg_{cnt}$, respectively, and so these attacks produce a new symbol. For unauthorised master device attacks, the model will fail to retrieve the DFA model according to $IP_{mst}$ and therefore raise anomalies. The model achieves over 99% precision and 100% F1-Score when detecting replay and unauthorised operation/device attacks. For DoS attacks, the proposed method achieves 99% recall because only severe DoS attacks can affect the operation pattern anomalies. The eavesdropping and tampering attacks, which increase the execution time, can be detected with 0.99% precision and 0.99% F1-Score. This evaluation result demonstrates that the proposed model is efficient in detecting ICPS attacks.

**Table 3.** The detection performance of the proposed model in seven attack scenarios.

| Attack Scenario | Precision | Recall | F1-Score | FPR(%) |
|---|---|---|---|---|
| Eavesdropping | 0.99 | 0.99 | 0.99 | 0.88 |
| Tampering | 0.99 | 0.99 | 0.99 | 0.97 |
| DoS | 0.99 | 0.99 | 0.98 | 0.83 |
| Selected forwarding | 0.99 | 0.99 | 0.98 | 0.92 |
| Replay | 1.00 | 1.00 | 1.00 | 0.70 |
| Unauthorised operation | 1.00 | 1.00 | 1.00 | 0.40 |
| Unauthorised device | 1.00 | 1.00 | 1.00 | 1.19 |

In tampering attack scenarios, attackers are able to tamper with any byte. The tampering position (e.g., function code, data) affects the detection performance. Their impact is summarised as follows.

- Suppose the tampered packets violate Modbus/TCP protocol. In this case, the first line can detect them as malformed request/response packets at the first defence line.
- If attackers tamper with $TI$ or $UI$, packets will fail in the reassembly phase. Consequently, these packets are classified as missing responses or unmatched responses at the first defence line.
- The modifications to $FC$, $Reg_{start}$ or $Reg_{cnt}$ affect the function type $FN$. They will violate the operation sequence pattern and thus be detected at the second defence line.
- Suppose attackers only tamper with response register data (e.g., temperature value), without affecting other fields. In this case, detecting them using the first two defence lines is difficult. However, tampering attacks, usually using MITM, increase the operation interval or response time. Therefore, the third or fourth defence lines are able to detect this case.

6.3.2. Performance of the Third Defence Line

The third defence line uses CKMeans to address the mix challenge and uses two methods (Section 4.2) to measure the time interval. The following section discusses how CKMeans helps to improve detection performance, followed by a discussion of the two measurement methods.

Figure 6 provides an example process for learning the time interval pattern for the transaction $q_1 \rightarrow q_2$. The distribution of the raw time interval values is shown in Figure 6a, which presents a big deviation of 51.03 ms. Furthermore, their probability density distribution shows that the values are significantly distributed in three parts. After using the CKMeans, the time interval values were spilt into three clusters, namely cluster #1, #2, and #3, as shown in Figure 6b. Then, we learnt the distribution pattern of each cluster using normal distribution models. As shown in Figure 6c–e, the learnt normal distribution models have relatively smaller deviations of 1.35, 1.39 and 1.31 ms. After using the CKMeans to partition values, the deviation dropped from 51.03 to an average of 1.35 ms, which is a significant improvement (39×). This result confirms the advantage of using CKMeans in learning the interval time distribution.

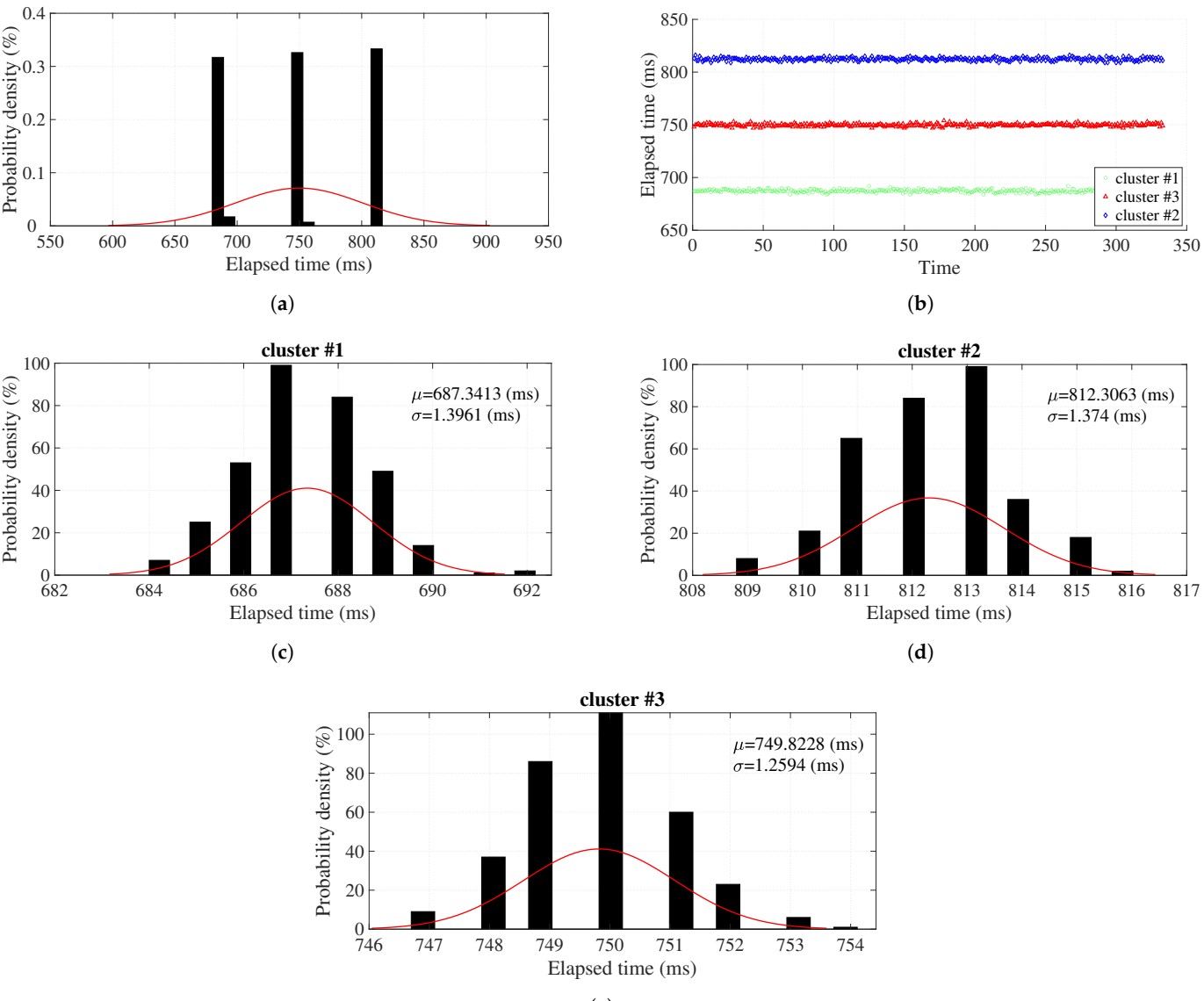

**Figure 6.** The process of learning the operation time interval pattern for the elapsed time $ET_{last}$. The $\sigma$ of the distribution is poor when directly learning the pattern on the original time interval values. After using CKMeans, these data were split into three clusters labelled with #1, #2 and #3. When learning distributions on each cluster, the $\sigma$ achieved an average of 1.35 ms, which obtains a significant improvement of 39 times compared to the original 51.03 ms. (**a**) The possibility distribution of $ET_{last}$ on original time interval values. The mean and deviation of this distribution are 760.242 and 51.03 ms, respectively. (**b**) The clustering result of CKMeans. (**c**) The possibility distribution of cluster #1. (**d**) The possibility distribution of cluster #2. (**e**) The possibility distribution of cluster #3.

This work further implemented two time-measure methods—$ET_{last}$ and $ET_{lastSim}$—to compare their impacts on detection performance. Table 4 shows the comparison results. The $ET_{last}$ achieved a good performance in precision, recall, and F1-Score. However, the $ET_{lastSim}$ performed worse on all evaluation metrics, e.g., 90% precision rate with a 77.07% false-positive rate. We further investigated the classification results of $ET_{lastSim}$ and realised that these results were caused by some normal operations being falsely classified as anomalous because $ET_{lastSim}$ is the sum of a set of corresponding $ET_{last}$. Therefore, if $ET_{last}$ is changed, it will affect all $ET_{lastSim}$ that contain this $ET_{last}$. Therefore, in the following evaluations, the $ET_{last}$ was used to measure the operation time interval.

**Table 4.** The evaluation results regarding how the time interval measurement methods and the time interval variation degree affect the detection performance of the third defence line.

| Measure Method | Deviation Degree (ms) | Precision | Recall | F1-Score | FPR (%) |
|:--:|:--:|:--:|:--:|:--:|:--:|
| $ET_{last}$ | [10,20) | 0.99 | 0.94 | 0.96 | 1.35 |
| | [20,50) | 1.00 | 1.00 | 1.00 | 0.62 |
| | [50,100) | 1.00 | 1.00 | 1.00 | 1.22 |
| $ET_{lastSim}$ | [10,20) | 0.90 | 0.72 | 0.77 | 76.54 |
| | [20,50) | 0.91 | 0.67 | 0.73 | 77.65 |
| | [50,100) | 0.91 | 0.68 | 0.75 | 77.03 |

### 6.3.3. Performance of the Last Defence Line

As shown in Table 2, the deviations in the operation execution time of this coal mine system were small; almost all of them are less than 2 ms. Therefore, the execution time defence line can accurately identify the anomaly operations whose execution time varies from the normal. As expected, and as shown in Table 5, this defence line achieved an average of 98.67% in F1-Score, with a 0.90% false-positive rate. Abnormal operations are easier to detect when the deviations become more severe.

**Table 5.** The evaluation results regarding how the time interval variation degree affects the detection performance of the operation execution time defence line.

| Deviation Degree (ms) | Precision | Recall | F1-Score | FPR(%) |
|:--:|:--:|:--:|:--:|:--:|
| $[10, 20)$ | 0.98 | 0.98 | 0.98 | 0.74 |
| $[20, 50)$ | 0.99 | 0.99 | 0.99 | 1.03 |
| $[50, 100]$ | 0.99 | 0.99 | 0.99 | 0.95 |

### 6.3.4. Interpretability Capability

As shown in Figure 7, the proposed model can provide detailed classification causes. These causes include the name of the defence line (red), the description of the normal pattern (blue) and the observed operation (purple). For example, the second report indicates that this operation violates the execution time defence line (fourth line) because the observed execution time is 94 ms, while the normal execution time range is $\mathcal{N}(58.66, 2.53^2)$, 94 greater than $58.66 + 2.53 * 3$. This is necessary for security experts to understand and quickly respond to ongoing attacks.

Unobserved state 10.41.12.85:4:1536:123 for operation {'IP_mst': '10.61.204.73', 'Addr_slv': '10.41.12.85', 'FC': 4, 'RN': 1536, 'Cnt': 123, 'ET': 124000, 'Rsp': 101.0}
Abnormal execution time 94.0, which not satisfied any of the following ND models: ['(58.65609348914858, 2.5270342171206366^2)']
Abnormal operation time interval 191000, which not satisfied any of the following ND models: ['(185000.0, 0.0^2)', '(187000.0, 0.0^2)', '(184000.0, 0.0^2)', '(186000.0, 0.0^2)', '(189000.0, 0.0^2)', '(188000.0, 0.0^2)', '(190000.0, 0.0^2)']

**Figure 7.** An example of a detection report for anomalous operations. The red parts indicate the name of the defence line, the blue parts are the observed value, and the purple parts are the description of the normal pattern.

### 6.4. Comparison Results

This section compares the proposed method with other state-of-the-art works regarding their detection capability and detection performance.

### 6.4.1. Detection Capability

The support of attack types, which usually refers to detection capability, is an important criterion for detection models. This work compared the detection ability with five state-of-the-art works, and the results are shown in Table 6. We further discuss these as follows:

- It is difficult to detect eavesdropping attacks and tampering attacks that do not affect the operation sequence and operation-related content using the operation sequence pattern proposed in [13,24,27,29,51]. By examining the time interval or execution time, our model can detect these at the third and fourth defence lines.
- The methods proposed in [13,24,27] used function code *FC* to represent the operation type. Therefore, they are limited in the detection of attacks that only tamper with the starting address $Reg_{start}$ and the number of operating registers $Reg_{cnt}$. This work is sensitive to these attacks by describing the function type as the combination of *FC*, $Reg_{start}$ and $Reg_{cnt}$.
- Attackers can launch a DoS attack via either traffic flooding (e.g., UDP, ICMP flooding) or malware programs (e.g., crypto-mining). Traffic flooding is easily detected using an IDS. However, detecting malware programs is challenging because they usually run on master devices and are absent in industrial traffic. The third and fourth defence lines are able to detect these by examining the operation time interval and response time.
- Most previous works [13,24,51] support detecting selective forwarding, replay, unauthorised operations and unauthorised devices, which usually change the operation sequence. This work proposed learning the operation pattern of each master–slaves channel rather than the master–slave channel. Therefore, this model has the ability to detect more sophisticated attacks.

**Table 6.** A comparison of the detection ability results of the proposed model and five state-of-the-art works.

| Method | Eavesdropping | Tampering | DoS | Selected Forwarding | Replay | Unauthorised Operation | Unauthorised Device |
|---|---|---|---|---|---|---|---|
| Proposed | ✓ | ✓ | ✔✔ | ✔✔ | ✔✔ | ✔✔ | ✔✔ |
| Operation sequence [13] | ✗ | † | ✓ | ✓ | ✓ | ✓ | ✓ |
| Critical state [29] | ✗ | † | ✓ | ✗ | ✗ | ✗ | ✗ |
| Temporal pattern [24] | ✗ | † | ✓ | ✓ | ✓ | ✓ | ✓ |
| Operation sequence [27] | ✗ | ✗ | ✓ | ✗ | ✗ | ✓ | ✗ |
| Outlier detection [51] | ✗ | ✗ | ✓ | ✗ | ✗ | ✗ | ✗ |

✓ supports, ✗ not supports, † limited supports, ✔✔ better supports.

### 6.4.2. Detection Performance

In this paper, four different approaches were further implemented and compared with the proposed method. These four approaches are:

- The first compared work was proposed by Kalech et al. [24]. They used five feature extraction methods to represent the operation pattern and used hidden Markov models (HMM) and artificial neural networks (ANN) to learn the pattern and detect anomalies.
- The second work uses the denoising autoencoder (DAE) and extreme gradient boosting (XGBoost) to build an anomaly detection and classification method.
- The other two works are implemented using *k*-means and principal component analysis (PCA). *k*-means is a popular algorithm in the area unsupervised anomaly detection, and clusters the dataset based on distance measurement and detects malicious operations by identifying outliers. The PCA is another unsupervised learning method, which identifies anomalies by examining objects' reconstruction errors.

The first work was implemented using the hmmlearn Python library, and the number of hidden states was set as four. TensorFlow and scikit-learn were used to implement the second work. The scikit-learn library was also used to implement the other two works. All the compared works were evaluated on real traffic collected from the coal mine factory and the simulated seven attacks were the same as the proposed work. The first compared work used its own features to illustrate the efficiency of the improved feature. The following three works used the same features as our model to evaluate whether the proposed model can compete with the complex DAE-XGBoost, *k*-means and PCA algorithms.

Table 7 represents the comparison results for the proposed model and four comparative approaches in seven attack scenarios. The proposed method achieved the highest performance among all compared works. The HMM-ANN [24] achieved the worst results because their features mainly focus on the time interval features; however, our model used the operation sequence, time interval and execution time features, which are more integrated. This result shows that only using the time interval feature is insufficient in detecting attacks and the improved features are more sensitive. The *k*-means and PCA models, which share the same features as our model, achieved a rather good detection performance. The DAE-XGBoost [52] achieved a better performance than the *k*-means and PCA. This demonstrates that the complex model (DAE-XGBoost is much more complex than *k*-means and PCA) could improve the detection performance. However, the proposed model achieved the best performance, although our model only used the basic techniques. These comparison results confirm the efficiency of the proposed model and suggest that the improved features could help the model better learn the operation pattern and basic techniques, as well as achieve a good detection performance by using more integrated features and carefully organising the detection model.

**Table 7.** Detection performance of the proposed model with four comparative approaches.

| Detection Method | Precision | Recall | F1-Score |
| --- | --- | --- | --- |
| Proposed | **1.00 *** | **1.00** | **1.00** |
| HMM-ANN [24] | 0.82 | 0.89 | 0.85 |
| DAE-XGBoost [52] | 0.93 | 0.93 | 0.90 |
| *k*-means | 0.91 | 0.92 | 0.90 |
| PCA | 0.92 | 0.92 | 0.89 |

* Bold means the best result.

## 7. Discussion

In the following, we discuss the insights derived from the experimental results and the limitations of this work.

### 7.1. Insights

In this study, we proposed an improved operation pattern and an unsupervised anomaly detection model. This model was evaluated with a coal mine ICPS and seven attack types and compared with nine state-of-the-art works. The evaluation and comparison results confirm that, by leveraging the improved features, the proposed model could also achieve a good performance and interpretability without requiring complex "black-box" algorithms. This work showed the importance of operation pattern features in anomaly detection models. This work suggests that future researchers should consider investigating the nature of ICPSs and using them to build interpretive models, rather than pursuing more complex ML and DL algorithms.

### 7.2. Proprietary Protocols

This work was conducted on the Modbus TCP protocol; however, we would like to clarify that the proposed model can be rapidly extended to other protocols. In the ICPS, most master and slave devices support open industrial control protocols (such as Modbus, opc ua [53]). However, manufacturers developed private protocols (e.g., Schneider's unified messaging application services (UMAS), Siemens' S7 Communication(S7comm)/S7comm Plus) and prefer to use these protocols in internal systems. These private protocols limited the proposed detection method because it needs to decode the ICPS traffic into plain text messages to extract operation features. Applying the proposed method to private protocols requires reverse engineering or the manufacturers' own efforts to decode the traffic to plain text.

### 7.3. Experimental Limitations

Due to the privacy of the ICPS infrastructure and production process, public ICPS collected from real ICPSs are rare on the Internet. This limited our method to only being tested on one real coal mine system. This work simulated cyberattacks relying on a testbed because it is not realistic to launch cyberattacks in working ICPSs. Because the ICPSs have strong constant and real-time features [4,8,11,15], we must assume that our model will work in other ICPSs. To promote further work, the dataset used in this work has been released.

### 8. Conclusions

In this paper, we investigated the operation pattern of a real coal mine system and improved the operation pattern features compared to previous works. Based on the improved features, this work combined the basic deterministic finite automaton (DFA) and normal distribution (ND) models to build an unsupervised anomaly detection model that uses a hierarchical structure to pursue interpretability. The evaluation results show that this model has a good detection performance of a 97% F1-Score. This model also achieved a 17% increase in precision and a 12% increase in F1-Score compared to previous works. These evaluation results confirmed the advantage of the proposed operation pattern features.

This work combined basic techniques to build the detection model; however, we would like to classify the advantages of the proposed model, as follows. (1) With improved features, the proposed model is able to achieve a better detection performance than complex models. This means that, for the same performance, the proposed model uses fewer computational resources and can meet the strict resource requirement of ICPSs. (2) The techniques used in this work are easy to interpret, which enables the proposed model to provide human-readable classification causes to help experts understand ongoing attacks. The black-box machine learning or deep learning algorithms often lack this interpretability. (3) The unsupervised techniques enable the proposed model to automatically learn the operation pattern without requiring labelled data. The realistic labelled data required by most ML or DL algorithms are difficult to collect because it is prohibited to launch attacks in real systems. These three advantages make the proposed model adaptable to more real systems, and it can be easily retrained if the operation patterns have been upgraded.

In conclusion, this work proposed a novel operation pattern-based anomaly detection model for ICPS, which uses improved features and combines basic techniques to detect attacks. The advantages of this model suggest that future researchers could focus on using the operation patterns to build intuitive, interpretable models rather than leveraging more complex machine learning or deep learning algorithms. For future works, we plan to test additional semi-supervised algorithms by labelling the class of some of the instances. We also plan to investigate the ability to learn the operation pattern from raw industrial traffic without requiring these patterns to be decoded into plain text request and response messages.

**Author Contributions:** Conceptualisation, methodology, software, validation, formal analysis, visualisation, writing—original draft preparation Z.C.; investigation, writing—review and editing, funding acquisition J.F.; resources, data curation, supervision, project administration B.C. All authors have read and agreed to the published version of the manuscript.

**Funding:** This work was supported by the National Natural Science Foundation of China under Grant 62001055.

**Institutional Review Board Statement:** Not applicable.

**Data Availability Statement:** Publicly available datasets were analysed in this study. This data can be found here https://github.com/WingPig99/ICPS-Dataset, accessed on 28 February 2023.

**Conflicts of Interest:** The authors declare no conflict of interest. The funders had no role in the design of the study; in the collection, analyses, or interpretation of data; in the writing of the manuscript; or in the decision to publish the results.

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
