# Peer review of "Rethinking the Operation Pattern for Anomaly Detection in Industrial Cyber–Physical Systems"

_applsci, doi:10.3390/app13053244_

Round 1

Reviewer 1 Report

The proposed paper named "Rethinking the Operation Pattern for Anomaly Detection in Industrial Cyber-Physical Systems" proposes new features new features to describe operation pattern to detect malicious behaviors or cyberattacks.
The paper is well organized with seven parts. The first part introduces the paper with and give some elements about the issues and the proposed solution. The second part realizes a state of the art about related works. The third part presents some preliminary elements about the Modbus protocol, with its different parts. The fourth part is the core of the work, and presents the operation pattern features, the data preprocessing and the detection model. The fifth part presents experiments and results, based on a coal mining factory. This part compares the proposed approach with different other approaches coming from the state of the art. The sixth part discuss about the proposed approach and its limitations. Finally, the seventh part concludes the paper. As this seventh part feels short, it may be merged with the previous part six.
The paper contains a lot of mistakes. For instance:
-    Cyberattack without space.
-    Line 49: ‘Main-in-the-Middle’ -> ‘man-in-the-middle’.
-    Line 86: ‘which makes it is more adaptable’
-    Line 87: ‘security experts understand”
-    Line 157: 'rotocol'
-    Etc.
Furthermore, the paper lack of explanations about results. For instance, line 484 to get the improvement of 3993%, which is very huge.

Reviewer 2 Report

The authors present a model for cyber-attack detection in industrial control systems using anomaly detection techniques. The proposed technique is applied to emulated coal mine traffic. The authors claim that their model improve existing work and their results are interpretable.

Overall, the article is well written and the topics covered are interesting. Cyber-physical systems and cyber-security are hot topics today.

However, this reviewer is not clear about the contribution of this paper. The authors present a technique based on four lines of defense using very basic techniques. In this reviewer's view, the proposed solution depends heavily on the traffic being periodic and constant. Interactions between masters and slaves should not vary over time. What are the implications of these constraints? Is this situation realistic in a real installation? Some discussion is needed.

The authors introduce cyber-physical systems and propose applying their model in detecting attacks on these types of systems. In cyber-physical systems, there is a strong inter-dependency between software and hardware components that is not apparent in the presented work, which is limited only to processing Modbus frames in conventional ICSs.

This reviewer misses a description of the elements that are part of the coal mine from which the traffic pattern shown in Table 2 is obtained. Similarly, it would be necessary to introduce how they interact. Is this pattern representative of a real installation? Is it always constant?

The model proposes the measurement of times between Modbus requests and responses. Specifically, it proposes measuring the execution time of a request by the slave. Where is this time measured? What are the implications of assuming this time constant? It is possible that some operations on the slave may require a variable time. Some discussion is needed.

Table 5 gives deviations ranging from 10ms to 100ms. How are these values obtained?

Some minor details:
Transmission time is not the same as round-trip time. In fact, sometimes the former is approximated as half the latter.

A language revision is needed.

Reviewer 3 Report

Dear Authors!
Thank you for submitting your article to MDPI! I found your article about Operation Patterns and Anomaly detection exciting and I recommend for publishing!
The number of references are enough, and the location and relevance is fine. The article contains 8 chapters, including the Introduction and Conclusion, which are well-edited and easy to read and follow. The pictures and tables are well-written and well-represented. The results in chapter 5  seams to be acceptable.

Before publication, I ask and recommend only 2 minor changes:
1, Please add a reference for "F1-Score" methodology
2, In line 157 I believe there is a typo "header and a rotocol data". Please correct!

Good luck!

Reviewer 4 Report

1.     Related work section is inadequate.

2.     Description about Operation Features and proposed model shall be separate sections.

3.     The authors have mentioned about the parameters they have considered / ignored in this section when discussing other related works. The parameters considered and ignored for their work may be summarized separately to understand the flow of the proposed work

4.     Section 3 gives a brief description of the Modbus TCP protocol. The title given here is "preliminary". No clarity in title.

5.     In section 3, Reason for ignoring the CRC value may be mentioned.

6.     In section 3 size of UI is not mentioned

7.     The authors may describe the ModBus TCP message format specifically used in the proposed work instead of mentioning the generic ModBus TCP/IP protocol. The authors may elaborate on the request and response results

8.     In section 4.2, The structure and the list of features after preprocessing may be summarized

9.     The methodology used in the first defense line to detect the attacks mentioned is not described

10.  The authors are ambiguous about the number of masters /slaves in their proposed work. No preliminary description of DFA, CK means normal distribution

11.  In section 4.3.2No detailed description about the parameters, states and size of the symbol. Example could be elaborated. Algorithm may be included

12.  In section 4.3.2No Values that represent the four categories of the symbol is not mentioned.

13.  In section 4.3.3 and 4.3.4 Algorithms may be included. I/O parameters is not clear.

14.  The parameters IPslv-The slave device IP address, Portslv -The TCP/UDP port and UI-The unit identity have been mentioned as the parameters used in Table 1  but they have not been mentioned in the table 2.

15.  In section 5 the authors may elaborate on how unauthorized operations and unauthorized devices are identified.

16.  The authors failed to mention if they have used the same dataset on all the other tested detection methods

17.  The values/outputs from the G-mean and their significance to support their proposed work is not discussed by the authors in section 5.

18.  The clarity in the overall assumptions in this work is missing.

19.  No sufficient illustrations (algorithms/flowcharts)  to support the proposed work.

10.  Evaluation criteria are not coherent which makes it difficult to understand the total number of factors that the authors have taken to support their results

Reviewer 5 Report

Paper deals with important task. The authors investigated traffic behaviour of a real coal mine system and proposed new features to describe operation pattern.

1. Related works should be extended using more papers similiar with conducted research. 

2. Section 4.1 should be extended using DOI10.1038/s41598-022-17254-4, DOI10.1038/s41598-022-11193-w

3. Conclusion section should be significantly extended.

4. Table 6 should contain the name of methods used for comparison. Also it would be good to analyse it in the Related works section. 

Modeling section is very good. It was a pleasure to read it.

Round 2

Reviewer 2 Report

In the opinion of this reviewer, the authors have made a great effort to improve the manuscript.

Some minor details:

- Please, unify the past participle of learn. Both "learned" and "learnt" are used.
- "This work first split" --> "splits"